# The Impact of the COVID Pandemic on the Incidence of Presentations with Cancer-Related Symptoms in Primary Care

**DOI:** 10.3390/cancers14215353

**Published:** 2022-10-30

**Authors:** Matthew P. Grant, Charles W. Helsper, Rebecca Stellato, Nicole van Erp, Kristel M. van Asselt, Pauline Slottje, Jean Muris, Daan Brandenbarg, Niek J. de Wit, Carla H. van Gils

**Affiliations:** 1Julius Centre for Health Sciences and Primary Care, University Medical Centre Utrecht, Utrecht University, 3584 CS Utrecht, The Netherlands; 2Department of General Practice, Amsterdam UMC Location University of Amsterdam, Meibergdreef 9, 1105 AZ Amsterdam, The Netherlands; 3Amsterdam Public Health Research Institute Program, 1081 BT Amsterdam, The Netherlands; 4Department of General Practice, Amsterdam UMC Location Vrije Universiteit, De Boelelaan 1117, 1081 HV Amsterdam, The Netherlands; 5Department of General Practice, Maastricht University Care and Public Health Research Institute, 6200 MD Maastricht, The Netherlands; 6Department of General Practice and Elderly Care Medicine, University Medical Centre Groningen, University of Groningen, 9712 CP Groningen, The Netherlands

**Keywords:** primary care, COVID, cancer diagnosis, care avoidance

## Abstract

**Simple Summary:**

The coronavirus pandemic profoundly affected how patients access health care services, as many individuals attempted to minimise risks of infectious contact and reduce burdens on health systems. This study aims to explore the effects of the coronavirus pandemic on patient presentations for cancer-related symptoms in primary care. It utilises routine clinical data for 1.23 million people in the Netherlands, comparing the first year of the pandemic to the two years prior. These data identify a 34% reduction in the incidence of cancer-related symptoms during the first wave (March to June 2020), with overall incidence returning to pre-corona levels after this period. In the first wave, the incidence of many symptoms was substantially reduced: breast lump (−17%), haematuria (−15%), abdominal mass (−21%), tiredness (−45%), lymphadenopathy (−25%), and naevus (−37%). In the second wave (October 2020 to February 2021), the incidence of breast lump and rectal bleeding was increased (both +14%), and tiredness was decreased (−20%), with the majority of other symptoms being similar to pre-COVID levels. These data describe large-scale primary care avoidance that did not increase until the end of the first COVID year for many cancer-related symptoms, suggestive that substantial numbers of patients delayed presenting to primary care.

**Abstract:**

**Introduction:** In the Netherlands, the onset of the coronavirus pandemic saw shifts in primary health service provision away from physical consultations, cancer-screening programs were temporarily halted, and government messaging focused on remaining at home. In March and April 2020, weekly cancer diagnoses decreased to 73% of their pre-COVID levels, and 39% for skin cancer. This study aims to explore the effect of the COVID pandemic on patient presentations for cancer-related symptoms in primary care in The Netherlands. **Methods:** Retrospective cohort study using routine clinical primary care data. Monthly incidences of patient presentations for cancer-related symptoms in five clinical databases in The Netherlands were analysed from March 2018 to February 2021. **Results:** Data demonstrated reductions in the incidence of cancer-related symptom presentations to primary care during the first COVID wave (March-June 2020) of −34% (95% CI: −43 to −23%) for all symptoms combined. In the second wave (October 2020–February 2021) there was no change in incidence observed (−8%, 95% CI −20% to 6%). Alarm-symptoms demonstrated decreases in incidence in the first wave with subsequent incidences that continued to rise in the second wave, such as: first wave: breast lump −17% (95% CI: −27 to −6%) and haematuria −15% (95% CI −24% to −6%); and second wave: rectal bleeding +14% (95% CI: 0 to 30%) and breast lump +14% (95% CI: 2 to 27%). Presentations of common non-alarm symptom such as tiredness and naevus demonstrated decreased in-cidences in the first wave of 45% (95% CI: −55% to −33%) and 37% (95% CI −47% to −25%). In the second wave, tiredness incidence was reduced by 20% (95% CI: −33% to −3%). Subgroup analy-sis did not demonstrate difference in incidence according to sex, age groups, comorbidity status, or previous history of cancer. **Conclusions:** These data describe large-scale primary care avoidance that did not increase until the end of the first COVID year for many cancer-related symptoms, suggestive that substantial numbers of patients delayed presenting to primary care. For those patients who had underlying cancer, this may have had impacted the cancer stage at diagnosis, treatment, and mortality.

## 1. Introduction

There is emerging evidence that the coronavirus (COVID) pandemic influenced many individuals to avoid accessing health care [1,2,3]. In The Netherlands, which was severely impacted by the first wave of the pandemic from March 2020, health care rapidly transitioned during this period, with the majority of primary care consultations shifting to telehealth, non-urgent appointments delayed or cancelled, and screening programs temporarily halted [4,5]. Normal health behaviours were also impacted by the pandemic, as government messaging and social expectations promoted ‘staying at home’, minimising contact, and to only seek health care for urgent matters [6,7,8]. This avoidance in accessing health care may have substantial and widespread implications, especially for conditions such as cancer, where over 80% of patients with cancer initially present to their General Practitioner (GP) [9,10]. Postponement of diagnosis and treatment can lead to later-stage diagnosis, worsening prognosis, increased health care costs and a greater impact on patient’s quality of life [1,10,11,12].

In The Netherlands, the COVID experience was comparable to many other European countries. March 2020 saw a rapidly escalating situation known as the first wave, culminating in a lockdown on 15 March [13]. These restrictions were reduced in June, and the country experienced a mostly unrestricted summer period with limited infections. Hospitalisations rose dramatically at the end of September 2020, signalling the start of the second wave, and subsequent lockdown from 14 October [13]. Health services remained under significant pressure with high numbers of infections and hospitalisations until March 2021, and corresponding tightening of restrictions during this period.

Cancer diagnosis rates were markedly reduced at the beginning of the COVID pandemic, with reductions between 25–39% reported internationally [14,15,16,17]. After these initial dramatic falls, cancer diagnoses in these countries returned to normal levels [15,17,18]. A number of studies described that reductions in diagnoses were largest for lower stage cancers [18,19,20]. However, diagnosis rates provide a limited context of the impact of COVID on cancer care, as they do not detail the causative factors, including potential health care avoidance or delays in diagnostic pathways. In the context of reported health care avoidance during the first wave of the COVID pandemic, it is unclear the extent and duration of this phenomenon for cancer care.

The diagnosis of cancer is a complex process reliant on multiple processes and individuals, working together to enable appropriate and timely investigation of patients [21]. The first step in this process occurs when the patient first experiences and acknowledges symptoms, and then seeks medical attention, also known as the ‘health seeking phase’ [7,22]. In the majority of countries, such as The Netherlands, the first contact occurs with a GP, who may pursue further investigations and referral based on the clinical circumstances [21,23]. Consequently, in such gatekeeper healthcare systems where the GP is the first point of access, the impact of COVID on health care usage is best measured in the primary care setting.

To assess the impact of the COVID pandemic on the cancer diagnostic pathway, we aim to detail the extent of consecutive phases of the COVID pandemic on primary health care presentations with cancer-related symptoms in The Netherlands.

## 2. Methods

### 2.1. Study Design

This study was designed as an observational cohort study incorporating routine clinical data from general practices throughout The Netherlands. For over 1.2 million primary care patients, these data are collected prospectively in the Intercity databases; a dynamic population cohort from five academic primary care networks throughout the Netherlands (Utrecht, Limburg, Groningen, and two in the Amsterdam region), including different metropolitan, rural and socio-economic regions. This study was reported in line with the Strengthening the Reporting of Observational Studies in Epidemiology (STROBE) Statement [24]. This research was reviewed by the institutional ethical review board of the UMC Utrecht (18-373/C) and considered not subject to the Medical Research Involving Human Subjects Act of the Netherlands.

The COVID period was defined as the period from 1 March 2020 onwards, informed by the Netherlands Institute of Health and the Environment (RIVM) [13,25]. We defined the first COVID period as between 1 March and 30 June 2020, and the second COVID period (incorporating the second and beginning of the third wave) from 1 October to 28 February 2021. The pre-COVID period was defined for the purpose of this study from 1 March 2018 to 29 February 2020, designed to provide two full years of data prior to the onset of COVID for comparison.

### 2.2. Population

All adult patients registered within the databases between 1 March 2018 and 28 February 2021 were included in the study. The Intercity databases comprise routine clinical care data collected from multiple general practice networks: the Academic Network of General Practice at Amsterdam UMC, location VU Medical Centre (ANH VUmc) and location Amsterdam Medical Centre (AHA AMC), Research Network Family Medicine (RNFM) Maastricht, the Academic General Practitioner Network Northern Netherlands (AHON) and the Julius General Practitioners’ Network (JGPN) databases [26]. In The Netherlands almost all patients are registered with a specific general practice clinic, with only those patients registered to the clinic included in the study.

### 2.3. Data Collection

Coded consultation data was collected for all participants, including demographics, and International Classification of Primary Care (ICPC-1) codes [27]. These ICPC codes are registered for each consultation, to describe the major presenting feature of the consultation. Demographics and patient characteristics include: ICPC codes for major comorbidities in the form of cardiovascular, diabetes, chronic obstructive airways disease; previous history of cancer; and, psychiatric or psychological diagnoses at any time point (see Appendix A). ICPC codes related to potentially cancer-related symptoms were identified in the literature by the researchers and supplemented and refined based on the clinical experiences of the primary care clinicians in the research group. Symptoms included are those with acknowledged associations with cancer, and relatively common occurrence (see Appendix A). Cancer alarm symptoms were defined as symptoms with a positive predictive value >5% for cancer according to literature, including: rectal bleeding, haematuria, dysphagia, post-menopausal bleeding, and breast lump [28,29].

ICPC codes for cancer-related symptoms were collected monthly for each adult patient from February 2018 to February 2021. The incidence of these symptoms was used as the primary outcome. The incidence of each of the codes was determined, with presentations for each ICPC code in a nine-month period counting as a single incidence of a potentially cancer-related presentation to primary care. Based on the clinical expertise of the research group, nine months was determined to be a clinically relevant period, in which presentations occurring after this time would likely require new diagnostic work-up, and thus should be considered a new incidence of that symptom [22].

### 2.4. Statistical Analysis

Incidence rates were calculated using the number of registered patients for each month as denominator. These were expressed at rates of per 100,000 people per month. Data from the two years prior to March 2020 served as a control, representative of pre-COVID patterns of cancer-related symptom incidence. Monthly incidences of cancer-related symptom presentations were analysed per time-period and expressed as a percentage of their pre-COVID incidence. In order to estimate incidence rate ratios (IRR), a negative binomial model was estimated for each symptom using the number of reported symptoms in a month as the outcome, period (pre-COVID, wave 1, summer, and wave 2) as the variable of interest and the natural logarithm of the number of registered patients in each month as an offset. In order to account for potentially non-linear time trends of symptom reporting over the duration of data collection, time in months since March 2018 was included as a natural cubic spline. Month of the year was included to control for seasonality in reporting of symptoms.

Extrapolation was used to predict differences in absolute incidence of symptom presentations for the entire population of The Netherlands, using the proportions from this study to predict national incidence for an adult population of 14.4 million people. Confidence intervals of 95% were employed. All analyses were conducted using SPSS (IBM, Chicago, IL, USA) version 26.

## 3. Results

### 3.1. Demographics

1,232,028 patients were included in the analysis described in Table 1. This population includes approximately 9% of the adult population of The Netherlands. 55% of the population were between 18 and 49 years, and 52% female.

### 3.2. Incidence of Cancer-Related Symptoms

Prior to COVID there were a mean 582 consultations per 100,000 population every month for new cancer-related symptoms. The incidence decreased by 34% (IRR 95% CI 0.57–0.77) to 404 per 100,000 during the first wave, and returned to normal levels (−2%, 529 per 100,000, CI 0.84–1.15) during the summer period and in the second wave (−8%, 575 per 100,000, CI 0.80–1.06), adjusted for monthly variability. Table 2 describes the pre-COVID incidence of cancer-related symptoms in primary care, and change in incidence over the first year of COVID as compared to previous years. Figure 1 represents the percentage change of major cancer-related symptoms over time for each month, comparing to the mean incidence for each month in the two years prior.

As shown in Table 2 and Figure 1, in the first wave, GP presentations for major cancer alarm symptoms were significantly decreased for breast lump (−17%, IRR 95% CI 0.73–0.94) and haematuria (−15%, CI −24% to −6%), but did not demonstrate significantly change for rectal bleeding (−13%, CI 0.76–1.00) and post-menopausal bleeding (−13%, CI 0.73–1.03). In the summer period, presentations were similar to pre-COVID incidence for these symptoms, apart from dysphagia that was increased (+25%, CI 1.07–1.46). In the second wave, the incidence of presentations was increased for breast lump (+14%, CI 1.02–1.27) and rectal bleeding (+14%, CI 1.00–1.30) and unchanged for haematuria (−1%, CI 0.90–1.10) and post-menopausal bleeding (−7%, 0.80–1.09).

As shown in Table 2 and Figure 1, presentations for other common cancer-related symptoms were significantly decreased during the first wave: for tiredness (−45%, IRR 95% CI 0.45–0.67), weight loss (−22%, CI 0.67–0.90), naevus (−37%, CI 0.53–0.75), and lymphadenopathy (−25%, CI 0.65–0.87). In the summer period, the incidence of symptoms did not exhibit differences in comparison to pre-COVID: for tiredness (−9%), naevus (+4%) and weight loss (+4%). The second wave demonstrated a significant reduction for tiredness (−20%, CI 0.67–0.97), with other symptoms demonstrating no variation: weight loss (−1%, CI 0.87–1.13), naevus (−4%, CI 0.81–1.13), and lymphadenopathy (−9%, CI 0.79–1.04).

### 3.3. Group Differences

Exploratory examination of the data revealed no clinically relevant differences between groups based on age, sex, major-comorbidities and history of cancer on the incidence of combined and specific cancer-related symptoms were observed. Figures illustrating this are provided in Appendix A.

### 3.4. Differences in Incidence over First Year of COVID

The dataset of 1,233,035 was extrapolated to The Netherlands adult population of 14.4 million people to represent how these changes in incidence may appear on a national scale, described in Table 3. Over the course of March 2020 to February 2021, as compared to the two years previous, there were increased incidence of presentations for rectal bleeding (+1509) and breast lump (+5751) in primary care. These additional presentations largely occurred from the summer period. There was a reduction in the incidence of haematuria in primary care of −1400 presentations. The incidence of tiredness (−100,883) and naevus (−40,337) were both substantially reduced over the first COVID year.

## 4. Discussion

### 4.1. Main Findings/Results of the Study

The results of this study demonstrate reductions in the incidence of most cancer-related symptoms to primary care during the first wave of COVID. These decreases during the first wave equate to thousands of patients who postponed or avoided attending primary care services during this period. In the summer period, as COVID infections and restrictions decreased, an apparent return to normal levels of incidence of cancer related symptoms was observed. After the first wave, an increase in the incidence of presentations for some cancer alarm symptoms was observed (see Figure 1 and Table 2). This may represent patients who delayed presentations throughout the earlier periods of the COVID pandemic. In contrast, whilst the incidence of non-alarm symptoms returned to normal levels after the first wave, an increase in incidence of presentations (which may represent delayed consultations) for these symptoms was not observed.

These data describe mass avoidance of primary healthcare during the first COVID wave for the majority of cancer-related symptoms. Whilst it is reassuring that over the course of the year presentations for these symptoms (especially cancer alarm symptoms) increased, this initial avoidance is cause for alarm. Sud et al. described the impact of diagnostic delay on cancer outcomes, a four month diagnostic delay is associated with worsening of 10 year mortality by 9%, 17%, and 6% for melanoma, colorectal, and breast cancer, respectively [11]. Based on the results for the alarm symptoms of rectal bleeding, breast lump and haematuria and using existing literature describing the positive predictive value of these symptoms of 5%, it would be expected that there were 131 patients who would go on to be diagnosed with breast cancer, and 137 with colorectal cancer who delayed presenting during the first wave [28]. These patients would have considerably worse outcomes, with their 10-year mortality rates decreased by 6–17%, based on a four-month delay [11]. Based on our data, we cannot determine the extent of this delay, however, an increase in the incidence of symptoms is not observed for most cancer-alarm symptoms until after the first wave, suggestive that delays in presentations to primary care may have been in excess of four months. These patients may be diagnosed at a later stage, undergo more intensive treatments, incur more health care costs and have greater disruption to their lives from their cancer diagnosis [30]. Modelling studies from Australia and Canada predict that these delays are likely to be associated with a 2% increase in cancer-related deaths, leading to an additional 940 deaths in The Netherlands [31,32].

For symptoms such naevus or tiredness, the degree of health care avoidance is even greater, with 40,000 and 100,000 fewer patients, respectively presenting nationally over the COVID year. This suggests potential selective avoidance, with a greater degree and length of avoidance in presenting to primary care in the case of less alarming cancer-related symptoms. Whilst these symptoms may be less alarming to many GPs, a recent study by White et al. found that the positive predictive value of tiredness for cancer was >3% for older adults [33]. A four-month delay is associated with 9% and 17% worse 10-year mortality for melanoma and colorectal cancers [31]. The incidence of both these symptoms normalised following the first wave, and then was again decreased during the second wave for tiredness. Neither of these symptoms exhibit an increase in presentations, suggesting that there are individuals who are yet to present with these symptoms or whose symptoms have resolved.

### 4.2. What This Study Adds

These results are consistent with Nicholson et al., whose study examined primary care cancer-related presentations in England in 2020, with a 24% reduction in presentations in 2020 compared to the year prior [16]. Similar patterns were observed with substantial reductions for all cancer-related symptoms at the beginning of the first wave, a period in the summer trending towards normalisation, and then a sudden, yet less severe decrease in the second wave [16]. The differences observed in this study between alarm and non-alarm symptoms were not described. Studies examining primary care use in The Netherlands reported that some 20–33% of patients avoided accessing primary care during the initial stages of the pandemic [3,34], consistent with our results demonstrating a 34% decrease in incidence of cancer-related symptoms.

The COVID pandemic remains a constant and evolving presence in our lives, influencing the provision of health care internationally. Whilst the dramatic impact and alterations of the first wave are unlikely to be repeated, it is apparent that for some symptoms, health care avoidance was not limited to this period and may be an ongoing issue throughout future fluctuations in the pandemic course. To facilitate timely diagnosis of symptoms, government policies should encourage primary care contact through future waves of COVID, in which many of these symptoms could be quickly triaged and assessed through telehealth. For specific alarm symptoms or those that require physical examination, physical consultation with the use of preventative measures can facilitate timely and correct diagnosis of these patients. However, the drivers of this avoidance are unclear and likely multifactorial. Further research would ideally seek to understand these reasons so that targeted education and policy interventions can be developed to address them at their root cause. Additionally, future research might supplement observation of cancer-related presentations to primary care with referral behaviour, since more selective GP presentation and GP referral behaviour could further expose the effects of health system impacts on potential delay in the diagnostic pathway and prognosis.

### 4.3. Strengths and Weaknesses/Limitations of the Study

This article describes the incidence of cancer related symptoms in a population of over 1.2 million people in The Netherlands during the first year of COVID, and data from the two years prior used as comparator. The population described comprised only of registered patients, and thus there was minimal variability of the cohort used as the denominator over the period. The incidence of new symptom presentations was used as the main outcome, so that one patient presenting on multiple occasions with the same symptom was counted as a single incidence. This is especially relevant given that changes to health service provision may have led to increased numbers of consultations for specific symptoms that require physical examination (i.e., rectal blood loss), but initial consultation may have occurred through telehealth.

The results of this study should be interpreted within an environment of rapid and large-scale changes to health systems throughout COVID. Normal care processes were altered, as a portion of care was transferred to telehealth, which may have limited diagnostic evaluation for symptoms that require physical investigation. The data employed in this study describes only primary care usage, and thus cannot detail how patient presentations for cancer related symptoms to other parts of the health care system were affected. Patients and health care providers may have been preoccupied with COVID infections, potentially resulting in less attention towards other symptoms, or leading to changes in coding.

The data is reliant on GP coding of these symptoms and does not consider the severity of these symptoms or clinician suspicion of an underlying cancer diagnosis. This reliance on GP coding of symptoms likely underestimates their prevalence in primary care, as many symptoms may be coded related to the diagnoses rather than the symptom (i.e., anal fissure rather than rectal bleeding). However, we have employed the data from the two years prior to act as control, which is coded by the same GPs for the same population, using the same approach. Telehealth and the focus on COVID diagnoses in primary may have additionally altered coding practices, but how and to what extent is unclear. The data demonstrated moderate month-to-month variability, which may be related to how public holidays and weekends were distributed, and this variability has been built into our analysis model.

## 5. Conclusions

This study demonstrates substantial decreases in the incident of presentations for cancer-related symptoms in primary care in The Netherlands at the beginning of the COVID pandemic. Increases in presentations for cancer alarm symptoms are observed at the end of 2020 and beginning of 2021, likely representing a ‘catch-up’ of patients who had delayed presenting during the earlier stages of the pandemic. A ‘catch-up’ in presentations was not evident for the majority of non-alarm symptoms, suggestive that these patients’ symptoms had either resolved or they were still yet to present. This data indicates that a substantial proportion of patients with cancer-related symptoms may have avoided or delayed visiting primary care, and that this has been an evolving situation throughout the COVID pandemic, not only being confined to the first wave. As described by Malagon et al., the full impact of COVID on delays in cancer diagnoses will likely take many years to be apparent [32]. As we enter a new and hopefully stable phase of the COVID pandemic, it is possible that health care avoidance is a continuing issue, that may have substantial impacts on patient care, mortality, and health expenditure for conditions such as cancer.

## Figures and Tables

**Figure 1 cancers-14-05353-f001:**
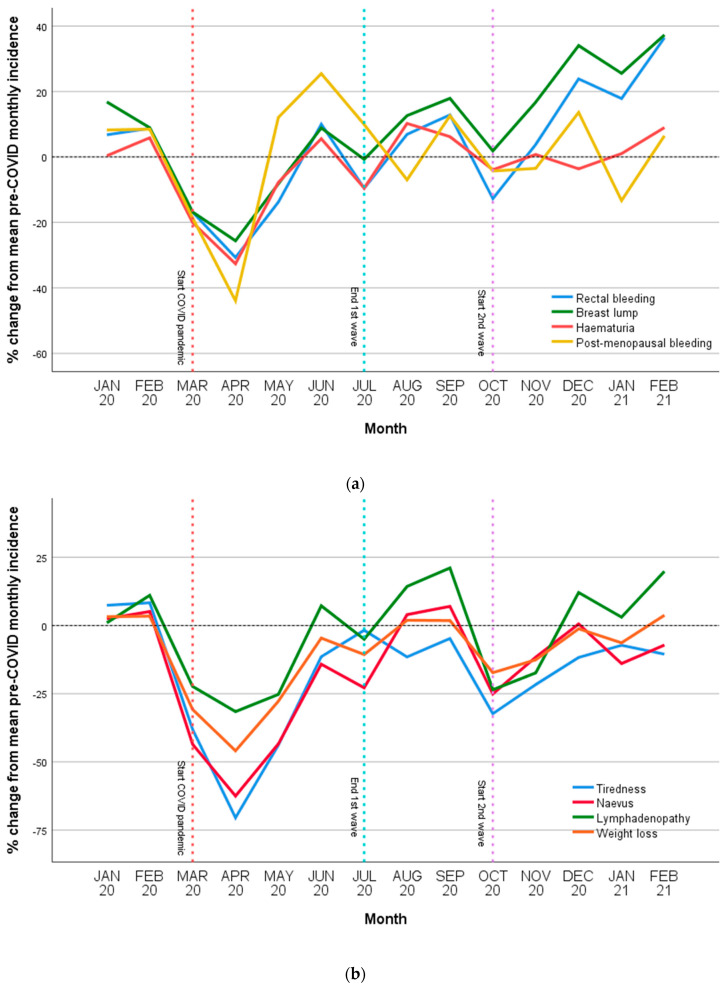
Percentage change in incidence over time of most prevalent cancer-related symptom presentations compared with their mean pre-COVID incidence for each month; (**a**), alarm symptoms (PPV > 5%), (**b**). non-alarm symptoms.

**Table 1 cancers-14-05353-t001:** Patient characteristics, as of February 2021.

		*n* (%)
Number patients included		1,233,035
Age Group	18–49	681,447 (55%)
	50–75	440,017 (36%)
	76 and older	111,571 (9%)
Sex	Female	637,242 (52%)
Major comorbidity	Cardiovascular Disease	128,796 (10%)
	Diabetes	71,665 (6%)
	Chronic obstructive airways disease (COPD)	14,768 (1%)
	Psychiatric/psychological	104,785 (9%)
History of cancer		46,779 (4%)

**Table 2 cancers-14-05353-t002:** Percentage change in incidence of cancer-related symptom presentations per time-period as a proportion of mean pre-COVID incidence. * includes all codes for cancer-related symptoms listed in the Appendix A.

	Mean Monthly pre-COVID Incidence	First Wave		Summer Period		Second Wave	
	Per 100,000 Population	Incidence Rate Ratio	95% CI	Incidence Rate Ratio	95% CI	Incidence Rate Ratio	95% CI
Rectal bleeding	163	0.87	0.76–1.00	1.04	0.90–1.21	1.14	1.00–1.30
Breast lump	190	0.83	0.73–0.94	1.02	0.90–1.16	1.14	1.02–1.27
Postmenopausal bleeding	69	0.87	0.73–1.03	0.99	0.82–1.19	0.93	0.80–1.09
Haematuria	120	0.85	0.76–0.94	1.01	0.89–1.13	0.99	0.90–1.10
Dysphagia	69	0.99	0.86–1.15	1.25	1.07–1.46	1.01	0.88–1.16
Abdominal mass	27	0.79	0.63–0.99	1.15	0.91–1.44	1.06	0.87–1.30
Melaena	23	0.78	0.61–0.99	1.27	0.99–1.63	1.18	0.96–1.46
Change in bowel habit	74	0.79	0.69–0.91	1.00	0.87–1.16	0.97	0.85–1.10
Tiredness	798	0.55	0.45–0.67	0.91	0.73–1.13	0.80	0.67–0.97
Lymphadenopathy	87	0.75	0.65–0.87	1.01	0.87–1.18	0.91	0.79–1.04
Naevus	379	0.63	0.53–0.75	1.04	0.86–1.25	0.96	0.81–1.13
Weight loss	124	0.78	0.67–0.90	1.04	0.90–1.21	0.99	0.87–1.13
All cancer symptoms combined *	582	0.66	0.57–0.77	0.98	0.84–1.15	0.92	0.80–1.06

**Table 3 cancers-14-05353-t003:** Estimated National differences in incidence of cancer-related symptoms over the first year of COVID adjusted for monthly variability. Only differences of >1000 are included.

	Mean Incidence per Month in The Netherlands Prior to COVID	Difference in Incidence 1st Wave	Difference in Incidence Summer Period	Difference in Incidence 2nd Wave	Annual Difference in Incidence March 2020 to Feb 2021 Compared to pre-COVID
Rectal bleeding	4578	−2333	+449	+3393	+1509
Breast lump	5354	−2108	+1514	+6345	+5751
Haematuria	3341	−1770	+195	+176	−1400
Change in bowel habit	2722	−1847	+351	+123	−1373
Tiredness	37,512	−63,582	−6600	−30,702	−100,883
Lymphadenopathy	3180	−2332	+875	−210	−1668
Naevus	17,239	−28,145	−3148	−9044	−40,337
Weight loss	4637	−4712	−359	−1533	−6605

## Data Availability

The de-identified data which were used for this analysis can conditionally be shared on reasonable request to the corresponding author, subject to the approval of the different datasources used.

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
