# Peer review of "The Impact of the COVID Pandemic on the Incidence of Presentations with Cancer-Related Symptoms in Primary Care"

_cancers, 2022, doi:10.3390/cancers14215353_

Round 1
Reviewer 1 Report
This paper is fine, though the subject area is now reasonably well understood. Therefore, some of my suggestions are on how we can learn from pandemic changes about other things that are relevant to cancer diagnosis.
Thus what is missing from this paper (from the interpretation/discussion mostly) is any sense of whether consultation behaviour changed over and above just numbers. I think it did, and the data supports my thinking. Let me explain...
The data (and other studies) make is clear there was a reduction in consultations for both low risk and for high risk cancer symptoms (mostly in wave 1). But (and this question could be answered by a further analysis) was this reduction in parallel with the general reduction in health care use? I suspect not, and my hypothesis is that as well as patients being unwilling to enter health care they were selective when they did enter. So was the fall in cancer symptom reporting larger, the same or smaller that the fall in consultations for other symptoms? My guess would be smaller - but it's only a guess, though could be answered. This is actually quite important, because it answers whether patients are passive recipients of healthcare, or whether they select what to report - and choose 'the more important things'. Of course they choose (lots of qualitative literature supports that - but can we provide some quantitative results for this? The smaller drops in breast lump and dysphagia consultations suggest this is true too. So, I'd like to see a benchmark of total consultations against which to benchmark the cancer symptoms data. I'd also like CIs on the graphs if possible. There's also a lot of overlap between the graphs and Table 2 (I prefer the graphs). It would additionally be helpful if we could see referral behaviour during the two periods. UK data suggests the fall in referral for possible cancer was smaller than the fall in GP cancer-possible consultations, suggesting GPs were being selective, just as patients were. I think a section in the discussion capturing this - or refuting it - would be good use of space.
I'm also uncertain about the concept of catch up consultations (though something must explain the rises in the second wave). If the patient has cancer, you'd expect the symptoms to persist or worsen, and these will be a catch up (or an emergency admission - are these easy to identify and report in this paper, or is it impossible?). But with a PPV of 5% 95% of symptoms will not be cancer, and many will disappear. So, there shouldn't be a large pool of unreported symptoms - instead there should be a small pool.
These two suggestions for improvement are my main ideas. There are some other strengths the team should be happy with. The ICPC is the best of the primary care coding systems for capturing symptoms (much better than Read/SNOMED for example, and so this study probably is the most robust of all of these. Furthermore, using the Sud data is a good strength (i'm always doing so!).
Author Response
Thank you for your review of our article. We have enclosed a document with our responses and changes in relation to your feedback.

Reviewer 2 Report
Reviewer report
Brief summary
Thank you for giving me the opportunity to review this manuscript, which explores the effect of the COVID-19 pandemic on the incidence of cancer-related symptoms reported to primary care in The Netherlands. Cancer is largely detected after patients start to develop symptoms, and doctors can only instigate cancer testing if people report their symptoms to a healthcare provider. Delayed symptom reporting is associated with diagnostic delay and poorer cancer outcomes. Quantifying delays in symptom reporting associated with the COVID-19 pandemic is important for a full understanding of COVID-19’s impact on cancer outcomes. The authors distinguish between symptoms that are: (1) alarming and relatively specific for a cancer site, with a positive predictive value for cancer of ≥5% (for example, breast lump for breast cancer), and (2) non-specific with lower positive predictive values for a range of cancers (e.g. fatigue, which is associated with leukaemia, mesothelioma, and cancers of the lung and ovary). This is an important distinction, as patients appraise the urgency of these two types of symptoms differently (possibly more so during the pressures of the pandemic), which affects their help-seeking behaviours. The study is set in a large and representative database of observational electronic health records, which enables the results to be generalised to The Netherlands. The incidence of all cancer-related symptoms recorded in primary care reduced by 31% (p<0.001) and by 5% (p<0.001), respectively, during the first (March-June 2020) and second (Oct 2020 to Feb 2021) waves compared with the pre-COVID mean incidence. The incidence of specific and alarming-symptoms (such as rectal bleeding and breast lump) decreased during the first wave, but recovered during the second wave. In contrast, the incidence of non-specific symptoms decreased in the first wave and remained lower than the pre-COVID mean incidence during the second wave. The authors suggest that substantial numbers of patients delayed presenting their symptoms of cancer to primary care during the pandemic and use existing models to anticipate the possible impact on cancer outcomes associated with COVID-19 in The Netherlands.
General concept comments
Article
My main comment relates to the authors’ choice of analyses, and I make some suggestions that they might like to consider. The Methods section would benefit for more detail on how the analyses were carried out. I think that the authors compared the incident rates at different time points using a Chi-squared test. There are several downsides to this approach, which does not maximise the benefits to be gained from the authors’ rich data source. The chi-squared approach only reports a p value, cannot be adjusted for potential confounders, and does not permit any estimation of uncertainty on the results. In addition, it does not adjust for seasonal variation in symptom occurrence. I understand that the number of degrees of freedom should be reported, and I could not find this information in the manuscript. I suggest that the authors consider using a negative binomial or Poisson regression approach, which would overcome these limitations and would strengthen the paper immensely. These methods are relatively straightforward to implement and interpret, and have been used by others to address this question in UK datasets (for example https://pubmed.ncbi.nlm.nih.gov/34031120/). I think the authors have visually inspected the data to check for differences by age, sex, comorbidities and history of cancer. Using negative binomial or Poisson regression would allow the authors to test these hypotheses more robustly.
Some of the conclusions drawn by the authors are wrong. For example, there is no evidence to conclude that, in the summer period, presentations were decreased for rectal bleeding (-7%, p=0.16) and post-menopausal bleeding (-4%, p=0.96) and increased for breast lump (+3%, p=0.22), or that they were increased in the second wave for post-menopausal bleeding (3%, p=0.15). I have marked other instances of this in the PDF.
The authors have erred towards over-interpreting their results. Their analyses do not model their data, but use chi-squared tests to test for differences. Any discussion of their “modelling” (e.g. page 6, line 208) is inappropriate.
Review
The Introduction generally covers the background well. Some of the studies cited (e.g. 4 and 5) in paragraph 1 of the Introduction were set in UK primary care. Please consider reframing this paragraph as suggesting that healthcare behaviours in The Netherlands may have changed during the pandemic, as suggested by behaviours elsewhere.
Specific comments are notes as follows:
Specifically, please check that the following reference citations are correct and appropriate:
1. Page 2, line 59: “After these initial dramatic falls, cancer diagnoses in most countries returned to normal levels in most counties (17).”
2. Page 2, line 68: “The first step in this process occurs when the patient first experiences and acknowledges symptoms, and then seeks medical attention, also known as the ‘health seeking phase’ (6).” Perhaps this reference would be relevant? https://pubmed.ncbi.nlm.nih.gov/22536840/
3. Page 3, line 89 “The COVID period was defined as the period from 1 March 2020 onwards, informed 88 by the Netherlands Institute of Health and the Environment (RIVM) (12).
Other specific comments
1. Page 3, lines 105-118. Out of interest, are coding systems other than ICPC-1 used in The Netherlands to record medical information in the electronic health record? For example, in the UK SNOMED-CT and Read codes are used.
2. Page 3, line 122. Please explain why a 9-month period was chosen to represent a single incidence of a potentially cancer-related symptom presented to primary care.
3. Page 3, lines 125-137. See above for my comments on the statistical analyses methods.
4. Page 3, Table 1. Please check the numbers: the sum of 681,447 + 440,017 + 111,571 is 123,035 and not 1,232,028.
5. Page 4, line 149. Please consider changing the text to “Table 2 [note incorrect table citation in the text] describes the pre-COVID incidence of a selection of cancer-related symptoms” to make the table clearer.
6. Page 4, Table 2. Please make it clearer to the reader that the total for all cancer symptoms combined includes symptoms not specified in the list above. It might be clearer to move the bottom line to the top, and then indicate that rectal bleeding down to weight loss represent a sample of the symptoms. Please also indicate why you picked out those particular symptoms to itemise in the table.
7. Page 4, lines 155-162 – see above for incorrect interpretation of some chi-squared test results.
8. Page 5, Figure 1. Can you include some pre-COVID data here, for example the Jan and Feb data, to help interpretation of the graph? It would be interesting to see how much of that initial change in symptom presentation occurred wholly in March 2020. (I would have expected the starting points for the graphs to be a 0% change)
9. Page 5, line 171 and line 173. From the p values, there is no evidence of a change in rate of presentation for weight loss and lymphadenopathy.
10. Page 6, lines 175-177. Group differences. Using your approach, I don’t think you can assert that there are no consistent differences between groups. Please also clarify what you mean by differences between groups here. It could be absolute difference at each time point, or differences in trends over time.
a. Supplementary Figure 1 suggests that there may be differences in absolute incidence by age group but without the 95%CI it is impossible to tell. The trends over time seem to be similar across the age groups.
b. Supp Fig 2 By sex, the absolute incidence and trends appear similar (95%CI would be helpful), apart from naevus, which may have a higher incidence in women than men.
c. Supp Fig 3 For comorbidity status, please clarify how this was defined. To be classified as "Yes", does a person have to have at least one diagnosis listed in the major comorbidity groups? There may be an absolute difference in incidence of rectal bleed between those with and without comorbidity (95%CI would help).
d. Supp Fig 4, agreed that the trends look similar, but absolute differences may exist for rectal bleeding.
11. Page 6, lines 179-180. Please clarify in the text that here you are extrapolating to national incidence over the first year of Covid, and cite Table 3. Please consider using modelling to estimate the differences at different time points with 95%CI, which would allow you to test whether there are actual differences here.
12. Page 6, Table 3. Please make it clearer where these data are discussed in the text by citing Table 3.
13. Page 6, Discussion. Please see above for comment relating to over-interpretation of the results.
14. Page 7, lines 221-224. “The most concerning aspect of these data is that avoidance was not only isolated to the first wave, and it is not until January and February that an increase in incidence is seen, and thus would expect that a substantial proportion of patients delayed seeking contact with primary care up to this point.” I am confused by these conclusions, because breast lump incidence seems to increase and stay above baseline from August 2020 (Figure 1a), and rectal bleed from Nov 2020. Apologies if I have misunderstood, but please would you clarify this argument as other readers may make the same mistake as me.
15. Page 8, lines 277-281. Please reconsider using the word "model" here, which isn't an appropriate term for a chi-squared test. If you opted to explore the negative binomial or Poisson modelling approach, you could make month-by-month comparisons. You could also include multiple presentations by individuals, by employing a random effect for patient.
16. Page 8, line 282. Please expand on the limitation of reliance on GP coding - for example, the likely underestimate of the numbers of people who have reported symptoms, and whether this underestimate is likely to be constant throughout the pandemic, or change with the different ways that primary care consultations take place.
Author Response
Thank you for your in-depth review of our article. We have enclosed a document with our responses and changes in relation to your feedback.

Reviewer 3 Report
This paper investigates the changes in the incidence reporting of cancer-related symptoms during the COVID-19 pandemic. While the topic is of interest from a public health point of view, I have some doubt about the rationale and the hypothesis of the paper. As the authors state in the Introduction, it is known that the cancer diagnosis rate dramatically fell during the first wave of the COVID pandemic, and it is not clear to me what is the added value of shifting the focus on incidence of cancer-related symptoms, that is a (weak) proxy for the hard outcome (cancer diagnosis).
Other issues:
1. The description of the source population is somewhat confusing. The authors state at first that this is a cohort study collecting data from general practices throughout the Netherlands, then it is state that the data were collected at five academic primary care center. If patients were recruited in academic primary care center, is this an issue with the external validity of the study?
2. At line 125, the authors state that "Incidence rates were calculated using the number of registered patients for each month at the denominator". The denominator of the incidence rate is time, not counts, so the authors should be more clear about this.
3. The threshold chosen for statistically significance in this paper is 0.01, which I think is sensible. However, an explanation for this choice should be provided.
Author Response

(The authors gave the same response as above.)

Round 2
Reviewer 2 Report
Thank you for considering all my suggestions, and for changing the analysis method. I have read your revised paper, and have made a few comments for your consideration in the PDF attached. They fall into two main categories:
1. Reference citations - I wonder if the tracked changes isn't displaying the citations properly, as some of the references in the Introduction seemed to be cited out of place. Please would you double check that?
2. Sometimes the text and tables don't match, so I've asked you to double check that.
3. I'm concerned about some overinterpretation of the data in places; for example, use of the phrase "non-significantly decreased". Where there is no evidence of a decrease, it's a bit slippery to suggest that there is one. I've flagged these in the attachment and perhaps you'd like to reconsider the wording?
Thanks for the opportunity to re-read your paper.
Best wishes
Sarah

Author Response
Thank you kindly for your in-depth review. Please find enclosed a table addressing your comments.
Kind regards
